# Multiple marginalized identities and internalized HIV stigma among people living with HIV in South Florida: An intersectional approach

Megan A. Jones[1], Elliott R. Weinstein[2,3], Steven A. Safren[3]*

1 Department of Prevention & Community Health, Milken Institute School of Public Health, George Washington University, Washington, District of Columbia, United States of America, 2 Behavioral Medicine Program, Massachusetts General Hospital, Boston, Massachusetts, United States of America, 3 Department of Psychology, University of Miami, Coral Gables, Florida, United States of America

☯ These authors contributed equally to this work.
* ssafren@miami.edu

## Abstract

Internalized HIV stigma is associated with several adverse mental and physical health outcomes among people living with HIV (PLWH). PLWH and other marginalized identities may experience worse internalized HIV stigma due to minority stress and structural oppression. This study conceptualized intersectionality via a multiplicative approach and examined the associations between intersectional marginalized-group identities and internalized HIV stigma among a sample of PLWH in South Florida ($N = 1343$) using hierarchical linear regression models. Interaction terms within these models were used to test the effects of having more than one marginalized identity over and above the main effects of each single marginalized identity. Overall, participants reported moderate levels of internalized HIV stigma ($M = 2.47$, $SD = 1.93$, range $1-6$) with main effects for those identifying with age, race, ethnicity, and/or gender marginalization experiencing higher levels. The interaction between gender and ethnic marginalization ($b = 0.82$) and the interaction between age and ethnic marginalization ($b = 0.32$), predicted significantly higher levels of internalized HIV stigma over and above the main effects of these variables on their own. Additionally, the interaction between age and racial marginalization ($b = -0.54$) and the interaction between age and sexual orientation marginalization ($b = -0.47$) both significantly predicted less internalized HIV stigma over and above the main effects of these variables on their own. Findings highlight the importance of considering how intersectional marginalized identifies influence PLWH's internalized stigma. Data offers insight into the subgroups of PLWH who could benefit from targeted interventions to reduce internalized HIV stigma and improve HIV care outcomes.

**Data availability statement:** We have made the de-identified dataset necessary to replicate this study available via Open Science Framework. The data can be accessed at the following URL: https://osf.io/xfte3/?view_only=35e6d-1c936a940d38361ea775237879b.

**Funding:** Data collection for this study was supported by the National Institute of Allergy and Infectious Disease (NIAID) (URL:https://www.niaid.nih.gov/) (P30AI073961 – PI Pawha) and the National Institute of Mental Health (NIMH) (URL:https://www.nimh.nih.gov/) (1P30MH133399 – PI Safren). Some author time was supported by the National Institute on Drug Abuse (NIDA) (URL: https://nida.nih.gov/) (R36DA058563-01A – PI Weinstein). The content is solely the responsibility of the authors and does not necessarily represent the official views of the National Institutes of Health. The funders had no role in study design, data collection and analysis, decision to publish, or preparation of the manuscript.

**Competing interests:** Dr. Safren receives royalties for authoring books published by Oxford University Press, Guilford Publications, and Springer/Human Press. The other two authors have no significant disclosures to report.

## Introduction

Despite advancements in prevention and treatment, HIV continues to be a major public health concern, both in the United States (U.S.) and globally. Although national HIV incidence rates in the U.S. have declined since 2018, certain subgroups, such as older adults and racial, ethnic, sexual, and gender minorities continue to experience HIV-related inequities [1]. For example, in 2019, approximately 71% of people living with HIV (PLWH) in the U.S. were people of color, despite making up less than 30% of the total population [2]. Similar trends hold for gay, bisexual, and other men who have sex with men (MSM) who reflect nearly 70% of HIV cases in the U.S. in 2019 [3] despite comprising less than 4% of the general population of men in the U.S. [4].

Multilevel HIV-related stigma and discrimination may perpetuate and exacerbate HIV-related inequities among marginalized groups. Beginning during the early stages of the HIV epidemic in the 1980s, PLWH have been victim to a wide range of discriminatory and prejudicial experiences such as social isolation [5], ostracism from social groups [5,6], and verbal and physical violence [7,8]. These experiences of discrimination result in several psychosocial challenges including a fear of disclosing one's HIV status [6,9], emotional distress [5], and avoidance coping [10]. Furthermore, PLWH who also identify with other marginalized identities (e.g., race, ethnicity, sexual orientation) may face even more complex patterns of stigma fueled by both misleading narratives surrounding the HIV epidemic and other systemic structures of inequity (e.g., racism, heterosexism, etc.) these individuals are burdened by [11,12]. For example, false narratives such as the idea that the "4 H's" (homosexuals, Haitians, heroin users, and hemophiliacs) were at fault for the epidemic [13], and initially referring to HIV as "GRID" (gay-related immune deficiency), suggesting that the disease is inherently related to homosexuality [14], have been pervasive since the start of the epidemic in the early 1980s, and still contribute to HIV-related stigma today, although they have long since been debunked. Still, these narratives have contributed to stigmas surrounding HIV that are still persistent today.

Several conceptual frameworks attempt to explain the mechanisms through which stigma perpetuates health inequities [15,16]; however, Earnshaw and Chaudoir's HIV Stigma Framework [17,18] is particularly effective in outlining how multiple levels of HIV stigma may impact PLWH. Although all three levels of stigma – enacted, anticipated, and internalized – directly impact health and well-being for PLWH, Earnshaw and Chaudoir point out that internalized stigma, or the degree to which PLWH sustain negative beliefs, feelings, and opinions about HIV in relation to themselves [13,15], may play an outsized role in these processes because of its strong influence on mental and behavioral health [14]. Specifically, PLWH who hold more internalized HIV stigma may have a harder time maintaining a positive outlook regarding their HIV status which can negatively impact their ability to engage in protective HIV-related health behaviors [18].

Internalized stigma has consistently been linked to poor HIV-related health outcomes. Studies domestically and internationally have observed that PLWH who hold greater levels of internalized stigma are more likely to have lower retention in the HIV care continuum [19,20], increased viral load [19,21], reduced access to HIV medical

care [22,23], and poorer antiretroviral therapy (ART) adherence [22,23]. PLWH who hold greater internalized HIV stigma also express difficulties receiving and integrating information about HIV medical care [24] and a lower likelihood of disclosing HIV status to sexual partners [25,26]. In addition to more negative physical health outcomes (e.g., poorer sleep quality, lower health-related quality of life) more generally [27,28], internalized HIV stigma is also associated with negative psychosocial outcomes such as more depressive and anxiety symptoms [10,29], hopelessness [30], dysfunctional coping mechanisms [10,31], loneliness and social isolation [32,33], problematic substance use [34], and increased sexual risk-taking [23,35].

PLWH and other minoritized identities often experience additional forms of stigma, that further reinforce HIV-related and general health inequities. This can be partially explained by the Minority Stress Theory, which posits that marginalized groups tend to experience more frequent physical health problems and psychological distress due to constant exposure to identity-based stigma and discrimination [36–38]. Although first conceptualized to highlight the unique stressors (e.g., discriminatory laws and policies, identity concealment, microaggressions) experienced by sexual minorities, Minority Stress Theory is now routinely applied to understand health inequities faced by individuals with other marginalized identities as well [37].

When considering exposure to prejudice or discrimination, PLWH who also identify as racial, ethnic, sexual, gender, and age minorities are subject to synergistic experiences of stigma, resulting from the pervasive structural systems of inequality and discrimination, that may exacerbate poorer HIV-related health outcomes compared to peers with no, or fewer, minoritized identities [11,39–41]. For example, prior research found that internalized HIV stigma was higher among Hispanic/Latina/x females, multiracial transgender individuals, and those under 50 years of age [42]. Similarly, a study conducted in China found that experiencing both HIV-related stigma and discrimination based on sexual orientation or gender led to poorer psychosocial functioning compared to those who only experienced HIV-related stigma [41]. Further, older adults living with HIV have been found to experience stigma related to both their HIV status and age, thus heightening discrimination, stereotyping, and feelings of rejection [39]. Other studies have found that intersectional stigma is a prominent part of life for midlife and older Black women living with HIV, who often experience negative responses to their race, gender, age, and HIV status, even among those who reported greater acceptance of their HIV status over time [43]. These intersecting health disparities demonstrate a need for further research to better understand how experiences of internalized HIV stigma differ depending on one's social positions and the various interlocking systems of oppression that impact their experiences as people with multiple minoritized identities.

Although marginalized-group races, ethnicities, ages, genders, and sexual orientations place individuals in unique social positions that, within the context of societal power dynamics, are often associated with a greater frequency of stigma-related experiences; this may not always be the case. It is possible that, some cases, a person's individual identifiers place them into social positions that may foster resilience from certain health outcomes [44–46]. For instance, there is mixed evidence that racially minoritized women living with HIV experience reduced internalized HIV stigma compared with their white peers [44]. Similarly, Emlet and colleagues [45] found that sexual minority men aged 55 years and older had significantly lower internalized HIV stigma than those 40 years and younger suggesting that age may serve as a protective factor against HIV-related stigma in certain subgroups of sexual minority men. Furthermore, individuals who identified as a sexual minority experienced less internalized HIV stigma compared with tan their heterosexual counterparts in a large sample of PLWH receiving care at a public HIV clinic in South Florida [46]. These mixed findings demonstrate a need for further research to tease out how these demographic factors may be associated with HIV-related stigma.

It is important to note that although a person's identities intersect at the individual level, they ultimately reflect several interconnected systems of privilege and oppression at the structural and societal levels [47]. Thus, Intersectionality [48] contributes an important theoretical framework that should be drawn upon to better understand the ways in which PLWH with multiple marginalized identities experience HIV-related stigma. Currently, there is little consensus on how to best measure intersectionality in behavioral health research quantitatively [49,50] with the majority of

 

studies exploring intersectionality qualitatively. One common practice employs an additive approach that suggests an increase in social inequality with each additional marginalized-group identity. Although the additive approach is easier to administer, it fails to adequately encapsulate the experiences of those with multiple marginalized identities [51]. Therefore, a multiplicative approach to measuring intersectionality is truly needed to demonstrate how outcomes, such as stigma, occur because of the interactions between identities [49–52]. Furthermore, few quantitative studies [41,52–54] have explored the relationship between marginalized-group identities and HIV-related stigma from an intersectional perspective with most of the published literature centering on qualitative approaches to this research question [40,55].

To the best of authors' knowledge, there are no quantitative studies directly examining the potential link between intersectional marginalized-group identities and internalized HIV stigma. This research question is of particular importance in South Florida, a racially and ethnically diverse region of the U.S. with significant HIV-related inequities. Thus, this current study aims to fill this important gap in the research by examining potential associations between intersectional marginalized-group identities and internalized HIV stigma within a low-resourced sample of PLWH who receive their HIV/AIDS care at a public hospital in South Florida. The specific aims of the study are to: (a) identify which, if any, marginalized identities are associated with internalized HIV stigma and [2] investigate whether there are significant interactions between marginalized identities that are associated with internalized HIV stigma.

## Methods

### Participants and procedures

Participants analyzed in this secondary analysis were recruited from a large HIV clinic within a public hospital system in South Florida. The data were collected from January 23, 2019, to February 17, 2023. Potentially eligible participants were approached by study staff members in the clinic waiting room and given the opportunity to enroll in the study if they met inclusion criteria. Participants were eligible for the study if they were 18 years or older, spoke English, Spanish, or Haitian Creole, received HIV care at this major hospital, and could provide informed consent. PLWH were excluded if they were currently incarcerated or unable to provide informed consent. This study was performed in line with the principles of the Declaration of Helsinki. Approval was granted by the University of Miami IRB 2016, IRB#20160911. Written informed consent was obtained from all participants prior to their participation in the original study.

### Measures

**Marginalized-group identity variables.** Participants were asked a series of demographic questions relating to identities including age, race, ethnicity, gender, and sexual orientation. These five variables were dichotomized such that dominant group identities served as the reference group while all other marginalized group identities were collapsed into the test group. [52]. Being under the age of 50 and self-identifying as White, non-Latino, cisgender male, and heterosexual were conceptualized as the reference or non-marginalized group for each demographic variable.

1. *Race:* Participants were asked to select the race(s) they self-identified as. PLWH were given the option to select White, Black/African American, Asian, Native Hawaiian/Pacific Islander, multiracial, or "another race" in the survey. Participants who selected a race other than White were a racial minority in this study and coded as 1. Participants who selected White were coded as 0 and served as the reference group for this variable.

2. *Ethnicity*: Participants were asked to select whether they self-identified as Hispanic/Latino/a/x. Participants who selected "yes" were coded as 1, and those who selected "no" were coded as 0 and served as the reference group.

3. *Age:* Participants were asked to report their age. Those who were aged 50 years or older were coded as 1, and those under 50 years old were coded as 0, serving as the reference group for this variable.

4. *Sexual orientation*: Participants were asked to self-report their sexual orientation. PLWH were given the option to select straight/heterosexual, gay or lesbian, bisexual, a different identity, or "I don't know". Participants who selected a sexual orientation other than straight/heterosexual were coded as 1. Participants who selected straight/heterosexual were coded as 0 and served as the reference group for this variable.

5. *Gender:* Participants were asked to select the gender with which they most closely identify with. The available options included cisgender male, cisgender female, transgender male, transgender female, or a different gender identity. PLWH who selected a gender other than cisgender male were coded as 1. Those who selected male as their gender identity were coded as 0 and served as the reference group.

## Internalized HIV stigma

Internalized HIV stigma was measured using the 6-item Internalized AIDS-Related Stigma Scale (Cronbach's alpha = 0.75) [56]. The scale consists of six items that assess participants' own HIV-related stigma. Participants reported whether they agreed (yes/no) with each statement with higher scores reflecting greater HIV-related internalized stigma. Example items on the scale included "being HIV positive makes me feel dirty" and "I hide my HIV status from others". Internalized HIV stigma is scored on a scale from 0–6.

## Statistical analyses

Descriptive statistics were examined to assess the distributions of the demographic and outcome variables. Next, bivariate linear regression models were used to assess the association between each individual marginalized-group identity and internalized HIV stigma. Then, a series of hierarchical linear regression models were employed to evaluate the associations between intersectional marginalized-group identities and internalized HIV stigma. Each model was composed of two blocks. The first block had one interaction term comprised of two marginalized-group identities (e.g., marginalized race and marginalized age; marginalized gender and marginalized ethnicity.). The second block included the interaction term and both main effects. As mentioned in the introduction, there is significant variation in how intersectionality is quantitatively evaluated in the literature. Some researchers advocate for analyzing interaction terms composed of two identifying factors (e.g., age and race) without their main effects included within the models while others demonstrate a utility of exploring interaction terms alongside their complementary main effects [50,57,58]. To be inclusive of both approaches, authors elected to employ this two-block approach such that block one included only the interaction while block two included both the main effects and the corresponding interaction term so to examine whether the interaction effects of multiple marginalization remained significantly related to internalized HIV stigma above and beyond the main effects of each social determinant of health alone. Alpha (α) was set to 0.05 for all analyses to indicate significance, and assumptions associated with simple linear regression (linearity, normality, independence, and homoscedasticity) were evaluated prior conducting the full regression analyses. All statistical analyses were conducted using SPSS version 26 [59].

## Results

The average age of participants was 49.9 years (*SD* = 11.9 years, range 21–80 years), with 59.1% of the sample being age 50 years or older. A majority (64.3%) of participants identified as Black/African American, and just under 34% identified as Hispanic/Latino/a/x. In addition, 59.6% of the sample identified as a cisgender male and 74.2% as heterosexual. The mean internalized HIV stigma score was 2.47 (*SD* = 1.92). Further descriptive statistics can be found in **Table 1**.

In the bivariate regression models, those who were 50 years or older experienced less internalized HIV stigma than those younger than 50 years by 0.4 units (b = −0.4, SE = 0.11, CI: −0.61, −0.19), and participants with a racial identity other than white reported less internalized HIV stigma compared to white participants by 0.32 units (b = −0.32, SE = 0.11, CI: −0.54, −0.10). Alternatively those who identified as a gender other than cisgender man experienced

**Table 1. Descriptive statistics for study sample (N = 1343).**

| Variable | Mean (SD) or *n* (%) |
|---|---|
| Age | 49.87 (11.93) |
| <50 years old[a] | 549 (40.88%) |
| 50 + years old | 794 (59.12%) |
| Race | |
| White[a] | 416 (31.02%) |
| Black/African American | 862 (64.28%) |
| Asian | 5 (0.37%) |
| Native Hawaiian/Pacific Islander | 3 (0.22%) |
| Native American | 4 (0.3%) |
| Multiracial | 23 (1.72%) |
| Other race | 28 (2.09%) |
| Ethnicity | |
| Not Hispanic/Latino/a/x[a] | 886 (65.97%) |
| Hispanic/Latino/a/x | 456 (33.95%) |
| Decline to answer | 1 (0.07%) |
| Gender | |
| Cisgender man[a] | 801 (59.64%) |
| Cisgender woman | 531 (39.54%) |
| Transgender man | 0 (0%) |
| Transgender woman | 10 (0.74%) |
| Different identity | 1 (0.07%) |
| Sexual Orientation | |
| Straight/heterosexual[a] | 997 (74.24%) |
| Gay or lesbian | 222 (16.53%) |
| Bisexual | 108 (8.04%) |
| Different identity | 9 (0.67%) |
| Don't know/decline to answer | 7 (0.52%) |
| Study outcome | |
| Internalized HIV stigma | 2.47 (1.92) |

[a]Represents the identities classified as the non-marginalized reference groups.

more internalized HIV stigma by 0.26 units (b = 0.26, SE = 0.11, CI: 0.05, 0.47) compared to those who identified as a cisgender man, and Hispanic/Latino/a/x participants experienced more internalized HIV stigma by 0.39 units (b = 0.39, SE = 0.11, CI: 0.17, 0.60) when compared with non-Hispanic/Latino/a/x participants. Sexual orientation was not statistically significant in the univariate linear regression model. All crude bivariate linear regression models are summarized in **Table 2**.

The interactive linear regression models are summarized in **Table 3**. In the hierarchical regression analyses with age and race categories, the interaction between older age and having a racial identity other than White was significant such that those who were aged 50 or older and non-white experienced significantly lower internalized HIV stigma by 0.54 units (b = −0.54, SE = 0.11, CI: [−0.75, −0.34], $p < .001$) compared to White participants under the age of 50 years. In the second block, which included both the interaction variable and the main effects, the interaction of age and race categories (older and non-White) was still significantly associated with less internalized HIV stigma (b = −0.59, SE = 0.23, CI: [−1.04, −0.15], $p = .01$), above and beyond the main effects of each marginalized identity alone.

Table 2. Crude bivariate linear regression analysis of marginalized-group identities on internalized HIV stigma.

| Variable | b (SE), [95% C.I.] |
|---|---|
| Age | |
| <50 years old | – |
| 50 + years old | −0.399 (0.106), [−0.608, −0.191]*** |
| Race | |
| White | – |
| Racial identity other than White | −0.322 (0.113), [−0.544, −0.100]** |
| Ethnicity | |
| Non-Hispanic/Latino/a/x | – |
| Hispanic/Latino/a/x | 0.388 (0.110), [0.171, 0.604]*** |
| Gender | |
| Cisgender man | – |
| Gender identity other than cisgender man | 0.260 (0.107), [0.050, 0.469]* |
| Sexual orientation | |
| Heterosexual | – |
| Sexual orientation other than heterosexual | −0.204 (0.120), [−0.440, 0.032] |

*$p$ < .05, **$p$ < .01, ***$p$ < .001.

Similarly, the interaction between age and sexual orientation categories (older than 50 and non-heterosexual) suggested that individuals who are aged 50 or older and have a sexual orientation other than heterosexual experienced significantly lower internalized HIV stigma by 0.47 units (b = −0.47, SE = 0.17, CI: [−0.79, −0.14], $p$ = .01) compared to heterosexual participants under the age of 50 years. However, this interaction was not significant in the second block when the main effects were also included.

Alternatively, the interaction between age and ethnicity categories (being 50 + years and Hispanic/Latino/a/x) was significantly associated with more internalized HIV stigma, such that participants who are aged 50 years or older and Hispanic/Latino/a/x experienced higher levels of internalized HIV stigma compared to non-Hispanic/Latino/a/x participants under the age of 50 years by 0.32 units (b = 0.32, SE = 0.14, CI: [0.05, 0.59], $p$ = .02). In the second block of the model, which included both the interaction term and the main effects, the interaction variable was still significantly associated with more internalized HIV stigma (b = 0.59, SE = 0.22, CI: [0.15, 1.03], $p$ = .01).

The interaction between gender and ethnicity categories (gender identity other than cisgender man and Hispanic/Latino/a/x) suggested that participants who did not identify as a cisgender man and who identify as Hispanic/Latino/a/x experienced greater internalized stigma than cisgender man, non-Hispanic/Latino/a/x participants by 0.824 units (b = 0.82, SE = 0.18, CI: [0.47, 1.18], $p$ < .001). In the second block of the model, when the main effects were included, the interaction trended towards significance (b = 0.45, SE = 0.18, CI: [−0.17, 0.92], $p$ = .06).

None of the other marginalized-group interactions were significantly associated with internalized HIV stigma.

## Discussion

This study is one of the first to quantitatively examine how intersecting marginalized-group identities may affect internalized HIV stigma among multiple marginalized PLWH in Miami, Florida, a U.S. HIV epicenter [60]. When examined individually, racial, ethnic, gender, and age minority identities were each associated with higher levels of internalized HIV stigma, while sexual orientation marginalization was not significantly related to internalized HIV stigma. This is consistent with prior research that suggests that there are higher levels of HIV-related stigma among marginalized groups [39,52,61].

**Table 3. Interactive hierarchical linear regressions models to examine associations between intersectional marginalized group identities and internalized HIV stigma.**

| Interaction Term | Block 1 (Interaction Term) b (SE), [95% C.I.] | Block 2 (Interaction Term Plus Main Effects) b (SE), [95% C.I.] |
|---|---|---|
| Age X Gender | 0.009 (0.126), [−0.238, 0.257] | 0.094 (0.216), [−0.330, 0.517] <br> −0.423 (0.139), [−0.696, −0.150]** <br> 0.180 (0.164), [−0.143, 0.502] |
| Age X Race | −0.541 (0.105), [−0.747, −0.335]*** | −0.594 (0.228), [−1.041, −0.148]** <br> 0.034 (0.187), [−0.332, 0.401] <br> 0.051 (0.167), [−0.277, 0.379] |
| Age X Ethnicity | 0.321 (0.138), [0.050, 0.592]* | 0.590 (0.222), [0.154, 1.026]** <br> −0.571 (0.133), [−0.831, −0.311]*** <br> 0.017 (0.166), [−0.308, 0.342] |
| Age X Sexual Orientation | −0.466 (0.166), [−0.793, −0.140]* | −0.024 (0.243), [−0.501, 0.453] <br> −0.442 (0.126), [−0.690, −0.194]*** <br> −0.285 (0.171), [−0.620, 0.049] |
| Gender X Race | 0.029 (0.114), [−0.194, 0.251] | −0.329 (0.240), [−0.801, 0.142] <br> 0.548 (0.204), [0.147, 0.949]** <br> −0.257 (0.140), [−0.532, 0.019] |
| Gender X Ethnicity | 0.822 (0.181), [0.466, 1.178]*** | 0.453 (0.239), [−0.016, 0.922] <br> 0.217 (0.128), [−0.034, 0.469] <br> 0.311 (0.137), [0.043, 0.579]* |
| Gender X Sexual Orientation | −0.337 (0.256), [−0.839, 0.166] | −0.517 (0.300), [−1.106, 0.072] <br> 0.311 (0.122), [0.073, 0.550]* <br> −0.018 (0.142), [−0.296, 0.260] |
| Race X Ethnicity | 0.243 (0.195), [−0.139, 0.625] | 0.097 (0.322), [−0.535, 0.729] <br> −0.138 (0.242), [−0.613, 0.336] <br> 0.269 (0.254), [−0.229, 0.768] |
| Race X Sexual Orientation | −0.160 (0.163), [−0.479, 0.159] | 0.637 (0.251), [0.144, 1.130]* <br> −0.617 (0.143), [−0.898, −0.337]*** <br> −0.686 (0.188), [−1.056, −0.317]*** |
| Ethnicity X Sexual Orientation | −0.109 (0.151), [−0.405, 0.188] | −0.486 (0.248), [−0.973, 0.000] <br> 0.616 (0.137), [0.348, 0.883]*** <br> −0.110 (0.169), [−0.442, 0.221] |

For block 2, the first line is the interaction term, the second line is the main effect for the first variable in the interaction, and the third line is the main effect for the second variable in the interaction. *$p < .05$, **$p < .01$, ***$p < .001$.

Additionally, the interaction between age and ethnic marginalized-group identities, as well as the interaction between gender and ethnic marginalized-group identities, serve as intersectional risk factors for internalized HIV stigma among PLWH. Hispanic/Latino/a/x participants who were age 50 years or older and Hispanic/Latino/a/x participants who self-reported a gender other than cisgender male reported greater internalized HIV stigma (0.32 and 0.82 units higher, respectively) compared to their White, non-Hispanic/Latino cisgender male peers. The data also suggest that the interaction between age and racial marginalized-group identities and the interaction between age and sexual orientation marginalized-group identities are protective factors for internalized HIV stigma such that older adults aged 50 years or older who have a racial identity other than White and/or have a sexual orientation other than heterosexual experience, on average, less internalized HIV stigma than their younger, heterosexual, and White counterparts.

Internalized HIV stigma was high among certain Hispanic/Latino/a/x populations, specifically, participants aged 50 years or older and those who identify as female and/or transgender. This finding is consistent with prior research examining the prevalence of HIV-related stigma among similar populations of Hispanic/Latino/a/x PLWH [42,62,63]. Several cultural factors, such as traditional female gender roles focused on sexual purity and passivity (*marianismo*), aggressiveness and social power of men over women (*machismo*), and traditional family values (*familismo*), may contribute to the greater levels of internalized HIV stigma among Hispanic/Latino/a/x women and transgender individuals [64–66]. Prior studies have found that Latina/x women who subscribe to these cultural ideals are less likely to disclose their HIV status to others, often resulting in higher rates of internalized HIV stigma [64–66]. These cultural ideals directed towards women and non-disclosure of HIV status may be partially responsible for the greater internalized HIV stigma among Latina/x women, but further research is needed to tease apart the complex relationship between ethnicity, gender, and HIV-related stigma.

Interestingly, although age appeared to be protective of internalized HIV among some demographic groups, internalized HIV stigma was higher among Hispanic/Latino/a/x participants aged 50 years or older compared to their younger peers. Although age tends to be associated with less internalized HIV stigma [45], findings in this study suggest that older adults' experiences of internalized HIV stigma vary. Prior research suggest that older adults' varying experiences with internalized HIV stigma may be due to factors such as cultural values, syndemic factors, and experiences with other types of structural stigma, and more research is needed to examine these hypotheses [67,68]. As researchers have described in previous studies [15,69], stigma resulting from the interlocking systems of oppression experienced by people with multiple marginalized identities perpetuates health disparities through mechanisms such as inequitable resource allocation, social isolation, maladaptive coping strategies, and stress. As such, in our study population, a diverse sample of largely under resourced PLWH with a high number of syndemic factors [46], similar structural inequities are likely to blame for the increased internalized HIV stigma in this group. In particular, older adults in the Hispanic/Latino/a/x community may face structural barriers due to immigration status and language barriers that function in tandem with cultural attitudes and religious beliefs that may both facilitate and exacerbate HIV-related stigma, internalized and enacted. In addition, the cultural ideals described above may also contribute to internalized HIV stigma among older adults who identify as Hispanic/Latino/a/x, as they may lead to a reduced willingness to openly discuss HIV and safe sex among one's peers [64], and, in turn, reduced social support [70]. Prior research suggests that social disintegration and a lack of social support may be pathways through which stigma is linked to adverse health outcomes, both general and HIV-related [15,67,71]. Overall, more research should be conducted to better understand the mechanisms that underlie these relationships between internalized HIV stigma and negative health outcomes among PLWH experiencing significant marginalization.

Unlike several other of the evaluated marginalized identities, age served as a protective factor against internalized HIV stigma in its interactions with both race and sexual orientation. PLWH over the age of 50 years who also identified as a sexual or racial minority reported significantly less internalized HIV stigma compared to their younger heterosexual and White peers, perhaps demonstrating a unique resilience that older sexual and racial minority PLWH possess. Past research is mixed regarding racial minorities' reporting of internalized stigma across the lifespan [20,72]; therefore, our finding that older racial minorities possessed less internalized HIV stigma is intriguing. It is possible that the racially minoritized older PLWH in our study experienced less internalized HIV stigma due to developing stigma competence (i.e., ability to successfully cope with experiences of stigma and discrimination) [73] associated with years of racism and HIV-related discrimination; however, more research must be conducted to systematically examine this hypothesis. Additionally, our finding that older sexual minority PLWH experience less internalized HIV stigma compared to their younger counterparts is consistent with past literature, and may be related to older sexual minority PLWH feeling a greater sense of "unit cohesion" [74], or a sense of community due to a shared history of social exclusion due to their HIV status [75,76]. Furthermore, it is possible that older racial and sexual minority PLWH generally hold less internalized HIV stigma because they are more likely to have been living with HIV for longer, allowing them more time to develop a sense of acceptance and peace about their diagnosis [77]. This is sometimes referred to as post-traumatic growth [78] and should be further

explored among PLWH and other marginalized identities, particularly in regions with significant HIV incidence and prevalence rates.

There are several limitations to the current study. First, because the analysis was cross-sectional, authors were unable to draw conclusions pertaining to causality or examine how internalized HIV stigma may change overtime based on intersecting marginalized identities. Second, the parent study only included a single measure of internalized HIV stigma, and thus we were not able to measure a broader range of HIV-related stigma (e.g., enacted, anticipated) or discrimination. Future studies should examine all three types of HIV-related stigma to create a more robust picture on how stigma may affect PLWH with additional marginalized identities. Third, the demographic variables included in the study were dichotomized to reflect a marginalized group and a non-marginalized group. This approach may have created challenges with capturing within-group differences (e.g., the gender marginalized-group identity variable included both cisgender women and transgender individuals; the racial marginalized-group identity variables included individuals who identified as Black/African American, Asian, Native American/Pacific Islander, and other races). Future research with larger sample sizes should more explicitly tease out these differences in sub-groups. Further, generalizability of the study is limited as all participants lived in South Florida; it is especially limited for transgender, non-binary, and other minority gender identities since these groups were not well represented in the analytic sample. Lastly, the analyses only captured the interactions between two marginalized-group identities at a time, and thus did not capture the experiences of those who are living at the intersection of three or more marginalized-group identities.

Despite its limitations, this study possesses several important strengths. First, the study population is relatively large (N = 1343) and diverse compared to many other studies that have examined factors associated with internalized HIV stigma. This large sample not only increases our power to detect meaningful differences between groups but also helps in generalizing findings for intersectional demographic groups that are often underrepresented in research. Our study population is also unique in that all participants lived in South Florida, a region with significant racial, ethnic, and sexual diversity and ample HIV-related inequities [60]. The study is also, to the authors' knowledge, the first quantitative study to use a multiplicative approach to intersectionality to directly examine the predictive capabilities of multiple marginalized identity variables in relation to internalized HIV stigma. Thus, the study makes an important contribution to the literature by identifying intersectional identities and the associated structural systems of inequality that are associated with internalized HIV stigma, informing future research and the development of stigma-reduction interventions to improve both general and HIV-related health outcomes for PLWH.

## Conclusion

With more than 80% of the study population reporting some level of internalized HIV stigma, HIV-related stigma is pervasive, and it is associated with several general and HIV-related adverse health outcomes [21,28,79]. Therefore, it is critical that interventions designed to support PLWH, and address to HIV-related stigma, adopt an intersectional approach to better address the needs of the specific subpopulations that face a greater burden of internalized HIV stigma. More specifically, there is a need for HIV stigma reduction interventions that are relevant and culturally appropriate for older adults, women, and transgender PLWH in Hispanic/Latino/a/x communities to help mitigate the harmful psychosocial effects of stigma and discrimination disproportionately experienced by these populations. Further research is needed to understand the interlocking systems of oppression that perpetuate stigma in varying populations and settings, thus informing the development of interventions to improve HIV care and quality of life outcomes among PLWH.

## Acknowledgments

We would like to thank Dr. Tamara Taggart and Dr. Deanna Kerrigan for their assistance with this project. Additionally, we would like to thank every participant in the study and the various study staff members who enrolled and interviewed participants.

## Author contributions

**Conceptualization:** Megan A. Jones.

**Data curation:** Steven A. Safren.

**Formal analysis:** Megan A. Jones, Elliott R. Weinstein.

**Funding acquisition:** Elliott R. Weinstein, Steven A. Safren.

**Investigation:** Steven A. Safren.

**Methodology:** Megan A. Jones, Elliott R. Weinstein.

**Project administration:** Megan A. Jones, Elliott R. Weinstein, Steven A. Safren.

**Writing – original draft:** Megan A. Jones, Elliott R. Weinstein.

**Writing – review & editing:** Megan A. Jones, Elliott R. Weinstein, Steven A. Safren.

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
