## [Decision Letter · Decision Letter 0]

27 Aug 2024

Dear Dr. Safren,

The study is relevant and timely. The findings have potential to inform the literature in this area. However, there are concerns about the intersectionality framework used in this study. Specifically, some of the arguments made in support of this framework appear to be in conflict with intersectionality. This may potentially skew the methods and findings. Please address these and other concerns raised by Reviewer 2 in both your Response to Reviewers and manuscript revisions.

We look forward to receiving your revised manuscript.

Kind regards,

Magdalena Szaflarski, PhD

Academic Editor

PLOS ONE

 [Data collection for this study was supported by the National Institute of Allergy and Infectious Disease (NIAID) (URL: https://www.niaid.nih.gov/) (P30AI073961 – PI Pawha) and the National Institute of Mental Health (NIMH) (URL: https://www.nimh.nih.gov/) (1P30MH133399 – PI Safren). Some author time was supported by the National Institute on Drug Abuse (NIDA) (URL: https://nida.nih.gov/) (R36DA058563-01A – PI Weinstein). The content is solely the responsibility of the authors and does not necessarily represent the official views of the National Institutes of Health.].  

3. In the online submission form, you indicated that [Data cannot be shared publicly due to concerns regarding confidentiality. Researchers can contact the senior author to request access to the data. All requests will go through the University of Miami IRB via proposed amendments, and researchers who meet the criteria for access to the confidential data will receive access via a secure, password- protected link to a University of Miami Box folder.]. 

Additional Editor Comments (if provided):

Reviewers' comments:

Reviewer's Responses to Questions

**Comments to the Author**

1. Is the manuscript technically sound, and do the data support the conclusions?

Reviewer #1: Yes

Reviewer #2: Partly

2. Has the statistical analysis been performed appropriately and rigorously?

Reviewer #1: Yes

Reviewer #2: No

3. Have the authors made all data underlying the findings in their manuscript fully available?

Reviewer #1: Yes

Reviewer #2: No

4. Is the manuscript presented in an intelligible fashion and written in standard English?

Reviewer #1: Yes

Reviewer #2: Yes

Reviewer #1: Very relevant research question with results and conclusions which will add significantly to current gaps in literature in HIV prevention. Especially excited to see the addition of age over 50 as SDOH as the epidemic continues to grow older. Excellent job. Just a few typos- ie pg 6 line 5 replace tan with than.

Reviewer #2: I would like to thank the authors and editor for the opportunity to review this interesting manuscript that explores some very important intersectional ideas. I have several concerns about the framing and methods that are outlined below. I hope that these will be taken as helpful and constructive as they are intended.

Introduction:

1. At the end of the second paragraph, the authors discuss misleading narratives about HIV and include, “(e.g., 4 H’s, GRID Disease).” Given the amount of time that has passed since these terms were used, and the standard practice of spelling out acronyms the first time they are used, these need to be spelled out. Further, simply spelling them out with no explanation may not convey what the authors are trying to say. Some may not have the historical knowledge to make sense of them. Lastly, the ‘D’ in GRID is disease so no need to say “Disease” after it.

2. While I applaud the use of intersectionality in this work, there are some ways in which the authors frame their arguments that are in conflict with the very tenets of intersectionality. For example, at the bottom of page 4, the authors state that PLWH experience “additional forms of stigma, unrelated to their HIV status,” and in the first sentence of the first full paragraph on page 5 that “minorities are subject to compounded experiences of stigma.” In an intersectional framing, those “additional forms of stigma” are not “unrelated to their HIV status.” One’s experience of racism or sexism is, in part, shaped by their HIV status and vice versa. For example, HIV stigma may take the shape of shaming a woman for being a bad mother because of her HIV status, or sexist framings of risk could blame her for being irresponsible as though her male partner had no role to play in her HIV acquisition. Second, the terms ‘compounded’ or ‘layered’ are also not consistent with intersectionality, as they imply that these experiences of stigma and oppression can be added up vs seeing them as co-constitutive and multiplicative.

3. At the bottom of page 5, the authors describe that “race, ethnicity, age, gender, and sexual orientation are traditional social determinants of health. It is essential that this be reframed. Race, itself, is not a SDOH; Racism is the SDOH. There is nothing inherent in one’s racial, ethnic, or sexual identity that determines health outcomes. It is the oppressions they are subjected to. This is also in conflict with intersectionality as intersectionality goes beyond identities and points to structure. This sentence goes on to state that “certain combinations of intersecting identities may foster resilience.” This, once again, centers identities, and I am not sure what this statement means.

a.This same reframing is needed in the third to last line on page 6 where the authors state that “outcomes, such as stigma, occur because of the interactions between identities

b.That identities are SDOH is restated in the first sentence under ‘Marginalized-group identity variables’ on page 8.

4. The very next sentence at the bottom of page 5 states that “greater racial diversity is associated with reduced internalized stigma,” but what does “greater racial diversity” mean? The magnitude of diversity is not an individual measure.

Methods

1. For the race category, I am confused as to why the authors decided to split the group into white and non-white when 95% of this sample is either Black or white. The “non-white” findings are likely reflecting the experiences of Black participants. If so, then this analysis and the findings need to be discussed specifically in that context. The authors have combined 862 Black participants with 63 participants that are divided into 5 other racial categories. It is unlikely that these data support making claims about all non-white people in this sample.

2. Similarly, nearly all of your participants were cis men/cis women. So, make your analysis about that. You don’t have enough Trans folks to make statements about them so you shouldn’t.

3. When it comes to Hispanic ethnicity, were there differences for Black H/NH vs White H/NH?

4. It is unclear why the authors chose to only include two-way interactions in the models. Given the use of intersectionality, it would be important to explore stigma among Black gay men, White heterosexual men, Black Hispanic lesbian women, etc. By treating these two-way interactions as discrete, as though they function independently of the other axes of identity that you have in your data undermines the intersectional analysis and weakens contribution of this framework.

Results

My only comment on the results is that they reflect the choices made on the methods as I describe above. I think they could be much stronger with a more robust intersectional analysis.

Discussion

1. The first sentence, once again, says that this study examined “intersecting social determinants of health.” But the authors measured identities not social determinants. This is repeated again in the third sentence.

2. In the third paragraph (page 17), the authors state, “findings in this study suggest that older adults’ experiences of internalized HIV stigma vary depending on a variety of cultural values, syndemic factors, and experiences with other types of structural stigma.” However, nothing in these findings support such a statement. There is no exploration of cultural values, syndemic factors, experiences with structural stigma in this study. In the very next sentence, the authors state again that stigma results “from intersecting identities.” (See comment#3 on the introduction.)

**Do you want your identity to be public for this peer review?** For information about this choice, including consent withdrawal, please see our Privacy Policy

Reviewer #1: **Yes: ** Pamela Payne Foster

Reviewer #2: No

---

## [Author Response · Author response to Decision Letter 1]

24 Oct 2024

Editor Comments

Thank you for submitting your manuscript to PLOS ONE. After careful consideration, we feel that it has merit but does not fully meet PLOS ONE’s publication criteria as it currently stands. Therefore, we invite you to submit a revised version of the manuscript that addresses the points raised during the review process.

The study is relevant and timely. The findings have potential to inform the literature in this area. However, there are concerns about the intersectionality framework used in this study. Specifically, some of the arguments made in support of this framework appear to be in conflict with intersectionality. This may potentially skew the methods and findings. Please address these and other concerns raised by Reviewer 2 in both your Response to Reviewers and manuscript revisions.

Thank you for this constructive feedback. We have addressed these concerns, as well as the other concerns raised by Reviewer 2, below. Based on this feedback, we have adjusted the framing of the paper to better align with intersectionality.

Reviewer 1

Very relevant research question with results and conclusions which will add significantly to current gaps in literature in HIV prevention. Especially excited to see the addition of age over 50 as SDOH as the epidemic continues to grow older. Excellent job. Just a few typos- ie pg 6 line 5 replace tan with than.

Thank you for your positive feedback. We have closely reviewed the manuscript and have corrected spelling, grammar, and syntax errors.

Reviewer 2

I would like to thank the authors and editor for the opportunity to review this interesting manuscript that explores some very important intersectional ideas. I have several concerns about the framing and methods that are outlined below. I hope that these will be taken as helpful and constructive as they are intended.

Thank you for your feedback and for the thoughtful suggestions to improve our paper.

At the end of the second paragraph, the authors discuss misleading narratives about HIV and include, “(e.g., 4 H’s, GRID Disease).” Given the amount of time that has passed since these terms were used, and the standard practice of spelling out acronyms the first time they are used, these need to be spelled out. Further, simply spelling them out with no explanation may not convey what the authors are trying to say. Some may not have the historical knowledge to make sense of them. Lastly, the ‘D’ in GRID is disease so no need to say “Disease” after it.

We have now spelled out these acronyms and added brief explanations to help contextualize the history of stigma in the HIV epidemic Please see the reframed language below:

“These false narratives – such as the idea that the “4 H’s” (homosexuals, Haitians, heroin users, and hemophiliacs) were at fault for the epidemic (13), and initially referring to HIV as “GRID” (gay-related immune disease), suggesting that the disease is inherently related to homosexuality (14) – have been pervasive since the start of the epidemic in the early 1980s, and still contribute to HIV-related stigma today, although they have long since been debunked.”

While I applaud the use of intersectionality in this work, there are some ways in which the authors frame their arguments that are in conflict with the very tenets of intersectionality. For example, at the bottom of page 4, the authors state that PLWH experience “additional forms of stigma, unrelated to their HIV status,” and in the first sentence of the first full paragraph on page 5 that “minorities are subject to compounded experiences of stigma.” In an intersectional framing, those “additional forms of stigma” are not “unrelated to their HIV status.” One’s experience of racism or sexism is, in part, shaped by their HIV status and vice versa. For example, HIV stigma may take the shape of shaming a woman for being a bad mother because of her HIV status, or sexist framings of risk could blame her for being irresponsible as though her male partner had no role to play in her HIV acquisition. Second, the terms ‘compounded’ or ‘layered’ are also not consistent with intersectionality, as they imply that these experiences of stigma and oppression can be added up vs seeing them as co-constitutive and multiplicative.

Thank you for this important feedback and for the suggestions on ways we can better apply the theory of intersectionality to this work. We have revised this paragraph to specifically highlight: a) the multiplicative nature of experiences of stigma and oppression and b) the intersecting nature of stigmas experienced by marginalized groups. Please see the reframed language below.

“PLWH and other minoritized identities often experience additional forms of stigma, that further reinforce HIV-related and general health inequities.”

At the bottom of page 5, the authors describe that “race, ethnicity, age, gender, and sexual orientation are traditional social determinants of health. It is essential that this be reframed. Race, itself, is not a SDOH; Racism is the SDOH. There is nothing inherent in one’s racial, ethnic, or sexual identity that determines health outcomes. It is the oppressions they are subjected to. This is also in conflict with intersectionality as intersectionality goes beyond identities and points to structure. This sentence goes on to state that “certain combinations of intersecting identities may foster resilience.” This, once again, centers identities, and I am not sure what this statement means.

In referring to race, ethnicity, age, gender, and sexual orientation as SDOH, we wanted to communicate that identities are what place an individual into the social context that generate “ism” related stigma, discrimination, and oppression. We appreciate the reviewers request to reframe this throughout our paper and have revised the introduction and discussion accordingly to highlight how a person’s individual identifiers place them into systems of oppression that contribute negatively to (or foster resiliency from) certain health outcomes. Please see below for the added language now included in the manuscript.

“When considering exposure to prejudice or discrimination, PLWH who also identify as racial, ethnic, sexual, gender, and age minorities are subject to synergistic experiences of stigma that may exacerbate poorer HIV-related health outcomes compared to peers with no, or fewer, minoritized identities.”

This same reframing is needed in the third to last line on page 6 where the authors state that “outcomes, such as stigma, occur because of the interactions between identities.”

We have rephrased this to indicate that these outcomes are the result of the social context and the intersecting systems of oppression experienced by people who hold multiple marginalized identities. Please see revised sentence below.

“Although marginalized-group races, ethnicities, ages, genders, and sexual orientations place individuals in unique social positions that, within the context of societal power dynamics, are often associated with a greater frequency of stigma-related experiences; this may not always be the case. It is possible that, some cases, a person’s individual identifiers place them into social positions that may foster resilience from certain health outcomes.”

That identities are SDOH is restated in the first sentence under ‘Marginalized-group identity variables’ on page 8.

This language has also been rephrased to state the following:

“Participants were asked a series of demographic questions relating to identities including age, race, ethnicity, gender, and sexual orientation.”

The very next sentence at the bottom of page 5 states that “greater racial diversity is associated with reduced internalized stigma,” but what does “greater racial diversity” mean? The magnitude of diversity is not an individual measure.

We have rephrased this sentence to state “For instance, there is some evidence that racially minoritized women living with HIV experience reduced internalized HIV stigma compared with their white peers.”

For the race category, I am confused as to why the authors decided to split the group into white and non-white when 95% of this sample is either Black or white. The “non-white” findings are likely reflecting the experiences of Black participants. If so, then this analysis and the findings need to be discussed specifically in that context. The authors have combined 862 Black participants with 63 participants that are divided into 5 other racial categories. It is unlikely that these data support making claims about all non-white people in this sample.

Consistent with prior research (Algarin et al., 2019), we chose to dichotomize all five identity variables (race, ethnicity, gender, age, and sexual orientation) to focus the analyses on internalized HIV stigma among marginalized-group identities (as opposed to non-marginalized group identities). While it would be interesting to examine more nuanced differences in internalized HIV stigma by identity-based variables (e.g., race, age, sexual orientation), that is beyond the scope of the current study. This is noted in the discussion/limitations section as a necessary direction for future research.

Similarly, nearly all of your participants were cis men/cis women. So, make your analysis about that. You don’t have enough Trans folks to make statements about them so you shouldn’t.

We agree that our sample does not include enough participants who identify outside the typical gender minority to meaningful evaluate the different between cisgender and transgender PLWH. Therefore, to evaluate the unique experience of non-gender dominant groups, we elected to lump together all participants who did not fall within the dominant gender identity “cisgender male”. We dichotomized the “gender” variable such that “1” represents participants who self-identified as either cisgender female, transgender male or female, non-binary, or another gender identity. The literature suggests that PWLH who identify with one of these gender identities often experience unique intersectional stigmas (Chakrapani et al., 2023; Rice et al., 2018). This approach has also been used in prior research (Algarin et al., 2019). We have now added some additional language and citations (Sangaramoorthy et al., 2017) to the introduction, methods, and discussion sections to highlight the reasoning behind this approach.

When it comes to Hispanic ethnicity, were there differences for Black H/NH vs White H/NH?

As mentioned in a prior response above, each of the five identity variables of interest, including race, was treated as a dichotomous variable for the analyses. Thus, we did not examine the individual nuanced differences in internalized HIV stigma between participants of different individual races. However, we did explore race X ethnicity such that internalized HIV stigma among white vs. non-white and among Hispanic vs. non-Hispanic participants was evaluated. Therefore, although we did not examine the unique experiences of Black Hispanic vs. non-Hispanic or White Hispanic vs. non-Hispanic participants, we did examine the differences in internalized HIV stigma among non-White, Hispanic participants compared to non-White, non-Hispanic participants.

It is unclear why the authors chose to only include two-way interactions in the models. Given the use of intersectionality, it would be important to explore stigma among Black gay men, White heterosexual men, Black Hispanic lesbian women, etc. By treating these two-way interactions as discrete, as though they function independently of the other axes of identity that you have in your data undermines the intersectional analysis and weakens contribution of this framework.

Thank you for this point. In writing the current paper, we hoped to begin to address a gap in the literature - since to our knowledge, there are no prior studies examining marginalized-group identities and internalized HIV stigma from an intersectional perspective. Researchers have historically utilized qualitative methods to examine intersectionality; therefore, ideas regarding the most appropriate quantitative methods to do so are continuing to emerge (Bowleg & Bauer, 2016). While analyzing three- and four-way interactions would be an important next step in truly evaluating intersectionality, we chose to include only two-way interactions in the models due to sample size challenges and the need to protect participant confidentiality. Unfortunately, with these challenges, there was little we could do with three- and four-way interactions in the current study. In addition, it is well-established in the literature that there are limitations to quantifying intersectionality; however, using the multiplicative approach to examine only the interactions, without the main effects in the model, is widely agreed upon to be the most appropriate method for quantitative studies seeking to take an intersectional approach (Bowleg & Bauer, 2016).

My only comment on the results is that they reflect the choices made on the methods as I describe above. I think they could be much stronger with a more robust intersectional analysis.

We appreciate this comment and agree with the reviewer that there are limitations in quantifying intersectionality, especially using traditional linear regression model approaches. Despite these limitations, utilizing a multiplicative approach to examine interactions between marginalized-group identity variables is a useful and appropriate quantitative method to represent the qualitatively unique experiences of multiple-marginalized populations. Future research should aim to improve our ability to quantify intersectionality, perhaps through machine learning methods like CART or random forests.

The first sentence, once again, says that this study examined “intersecting social determinants of health.” But the authors measured identities not social determinants. This is repeated again in the third sentence.

We have revised these sentences based on the reviewers’ feedback around the framing of the paper. Please see the revised language below:

“This study is one of the first to quantitatively examine how intersecting marginalized-group identities may affect internalized HIV stigma among multiple marginalized PLWH in Miami, Florida, a U.S. HIV epicenter.”

In the third paragraph (page 17), the authors state, “findings in this study suggest that older adults’ experiences of internalized HIV stigma vary depending on a variety of cultural values, syndemic factors, and experiences with other types of structural stigma.” However, nothing in these findings support such a statement. There is no exploration of cultural values, syndemic factors, experiences with structural stigma in this study. In the very next sentence, the authors state again that stigma results “from intersecting identities.” (See comment#3 on the introduction.)

We have rephrased the language to clearly indicate that the findings of our study suggest that older adults’ experiences of internalized HIV stigma vary, and according to the literature, this may be due to factors like cultural or generational values, syndemic factors, and other types of structural stigma. In addition, we have added language to the next sentence to emphasize that stigma is not caused by identities themselves, but rather, is a result of the social context surrounding such identities and have reframed our manuscript with this in mind. Please see below for revisions.

“Although age tends to be associated with less internalized HIV stigma (45), findings in this study suggest that older adults’ experiences of internalized HIV stigma vary. Prior research suggest that these varying experiences with internalized HIV stigma may be due to factors such as cultural values, syndemic factors, and experiences with other types of structural stigma, (67,68). As researchers have described in previous studies (15,69), stigma resulting from the interlocking systems of oppression experienced by people with multiple marginalized identities perpetuates health disparities through mechanisms such as inequitable resource allocation, social isolation, maladaptive coping strategies, and stress.”

---

## [Decision Letter · Decision Letter 1]

2 Jun 2025

Dear Dr. Safren,

Thank you for submitting your manuscript to PLOS ONE. After careful consideration, we feel that it has merit but does not fully meet PLOS ONE’s publication criteria as it currently stands. Therefore, we invite you to submit a revised version of the manuscript that addresses the points raised during the review process.

While the authors have made commendable efforts in addressing the reviewers’ comments, a few minor issues remain, please address the following points:

Reframe language that implies stigma results from individuals' identities, instead, emphasize the structural systems (e.g., racism, sexism) that produce stigma.Replace references to identities (e.g., race, gender) as social determinants of health (SDOH) with references to the structural forces that shape health outcomes.Provide a clearer justification for the grouping of diverse racial or gender identity categories (e.g., “non-white” or “non-cis male”).Discuss the potential implications of combining heterogeneous participant groups on the interpretation of findings.Ensure all reviewer comments are explicitly and thoroughly addressed in both the revised manuscript and the response letter. In particular, revisit any areas where concerns were acknowledged but not fully incorporated into the manuscript.

Could you please carefully revise the manuscript to address all comments raised?

We look forward to receiving your revised manuscript.

Kind regards,

Zahra Al-Khateeb, Ph.D

Staff Editor

PLOS ONE

Journal Requirements:

Reviewers' comments:

Reviewer's Responses to Questions

**Comments to the Author**

Reviewer #1: All comments have been addressed

2. Is the manuscript technically sound, and do the data support the conclusions?

Reviewer #1: Yes

3. Has the statistical analysis been performed appropriately and rigorously?

Reviewer #1: Yes

4. Have the authors made all data underlying the findings in their manuscript fully available?

Reviewer #1: Yes

5. Is the manuscript presented in an intelligible fashion and written in standard English?

Reviewer #1: Yes

Reviewer #1: This is a revision review. The authors provided adequate revisions or through explanations to support their perspectives. These revisions meet the criteria for publication.

**Do you want your identity to be public for this peer review?** For information about this choice, including consent withdrawal, please see our Privacy Policy

Reviewer #1: **Yes: ** Pamela H Foster

---

## [Author Response · Author response to Decision Letter 2]

11 Jul 2025

Editor Comments

Thank you for submitting your manuscript to PLOS ONE. After careful consideration, we feel that it has merit but does not fully meet PLOS ONE’s publication criteria as it currently stands. Therefore, we invite you to submit a revised version of the manuscript that addresses the points raised during the review process.

Thank you for this helpful feedback. We have addressed these concerns below.

While the authors have made commendable efforts in addressing the reviewers’ comments, a few minor issues remain, please address the following points:

Reframe language that implies stigma results from individuals' identities, instead, emphasize the structural systems (e.g., racism, sexism) that produce stigma.

We have further clarified this by adding language to emphasize that stigma results from structural systems of inequity such as racism, sexism, and heterosexism, rather than from individuals’ marginalized identities. Please see an example of the reframed language below that can be found throughout the paper:

“Furthermore, PLWH who also identify with other marginalized identities (e.g., race, ethnicity, sexual orientation) may face even more complex patterns of stigma fueled by both misleading narratives surrounding the HIV epidemic and other systemic structures of inequity (e.g., racism, heterosexism, etc.) these individuals are burdened by.”

Replace references to identities (e.g., race, gender) as social determinants of health (SDOH) with references to the structural forces that shape health outcomes.

Thank you for this important feedback. We have removed all references to identities as social determinants of health and replaced these with references to the structural systems of inequality that shape health outcomes. Please see the examples below on pages 19 and 21.

“Unlike several other of the evaluated marginalized identities, age served as a protective factor against internalized HIV stigma in its interactions with both race and sexual orientation.” (page 19)

“Thus, the study makes an important contribution to the literature by identifying intersectional identities and the associated structural systems of inequality that are associated with internalized HIV stigma, informing future research and the development of stigma-reduction interventions to improve both general and HIV-related health outcomes for PLWH.” (page 21)

Provide a clearer justification for the grouping of diverse racial or gender identity categories (e.g., “non-white” or “non-cis male”).

Consistent with prior research (Algarin et al., 2019), we chose to dichotomize all five identity variables (race, ethnicity, gender, age, and sexual orientation) to focus the analyses on internalized HIV stigma among marginalized-group identities (as opposed to non-marginalized group identities). A significant reason behind this choice was that breaking down the study sample into more specific intersectional groups to examine sub-group differences would create very small sample sizes for some of the groups (e.g., Asian transgender men), leading to decreased statistical power that would negatively affect our ability to identify significant findings. In addition, this may also create confidentiality issues due to the very small sample sizes for some of the intersectional groups.

Further, have also rephrased language throughout the manuscript to more clearly differentiate between the marginalized-group identities and non-marginalized group identities for each of the five identity variables (e.g., “white” and “racial identities other than white”, “heterosexual” and “sexual identity other than heterosexual”, “cisgender man” and “gender identity other than cisgender man”.

Discuss the potential implications of combining heterogeneous participant groups on the interpretation of findings.

This is discussed in the limitations section of the manuscript. Please see an example of the language below:

“Third, the demographic variables included in the study were dichotomized to reflect a marginalized group and a non-marginalized group. This approach may have created challenges with capturing within-group differences (e.g., the gender marginalized-group identity variable included both cisgender women and transgender individuals; the racial marginalized-group identity variables included individuals who identified as Black/African American, Asian, Native American/Pacific Islander, and other races). Future research with larger sample sizes should more explicitly tease out these differences in sub-groups.”

Ensure all reviewer comments are explicitly and thoroughly addressed in both the revised manuscript and the response letter. In particular, revisit any areas where concerns were acknowledged but not fully incorporated into the manuscript.

We have gone through once again, thoroughly, to make sure that all of the concerns have been incorporated into the manuscript, per some of the comments above this has been generally the case. We discovered one area where a concern was acknowledged, but not incorporated into the manuscript: the inclusion of only two-way interactions in the models and not three- or four-way interactions. Upon consultation with colleagues with significant expertise in this space, authors chose not to include additional interactions because with this sample size, three- and four-way interactions would not likely produce meaningful results.

As described in our previous revision, in writing the current paper, we hoped to begin to address a gap in the literature - since to our knowledge, there are no prior studies examining marginalized-group identities and internalized HIV stigma from an intersectional perspective. Researchers have historically utilized qualitative methods to examine intersectionality; therefore, ideas regarding the most appropriate quantitative methods to do so are continuing to emerge (Bowleg & Bauer, 2016). While analyzing three- and four-way interactions would be an important next step in truly evaluating intersectionality, we chose to include only two-way interactions in the models due to sample size challenges and the need to protect participant confidentiality. Unfortunately, with these challenges, there was little we could do with three- and four-way interactions in the current study. In addition, it is well-established in the literature that there are limitations to quantifying intersectionality; however, using the multiplicative approach to examine only the interactions, without the main effects in the model, is widely agreed upon to be the most appropriate method for quantitative studies seeking to take an intersectional approach (Bowleg & Bauer, 2016).

---

## [Editor Report · Decision Letter 2]

24 Jul 2025

Multiple-marginalized identities and internalized HIV stigma among people living with HIV in South Florida: an intersectional approach

PONE-D-24-23097R2

Dear Dr. Safren,

We’re pleased to inform you that your manuscript has been judged scientifically suitable for publication and will be formally accepted for publication once it meets all outstanding technical requirements.

Kind regards,

Laura Kelly, PhD

Division Editor

PLOS One